# Convolutional Neural Networks for Hole Inspection in Aerospace Systems

**DOI:** 10.3390/s25185921

**Published:** 2025-09-22

**Authors:** Garrett Madison, Grayson Michael Griser, Gage Truelson, Cole Farris, Christopher Lee Colaw, Yildirim Hurmuzlu

**Affiliations:** Lyle School of Engineering, Southern Methodist University, Dallas, TX 75205, USA; ggriser@smu.edu (G.M.G.); gtruelson@smu.edu (G.T.); cfarris@smu.edu (C.F.); ccolaw@smu.edu (C.L.C.); hurmuzlu@smu.edu (Y.H.)

**Keywords:** aerospace inspection, FOd detection, computer vision, convolutional neural networks, embedded imaging

## Abstract

Foreign object debris (FOd) in rivet holes, machined holes, and fastener sites poses a critical risk to aerospace manufacturing, where current inspections rely on manual visual checks with flashlights and mirrors. These methods are slow, fatiguing, and prone to error. This work introduces HANNDI, a compact handheld inspection device that integrates controlled optics, illumination, and onboard deep learning for rapid and reliable inspection directly on the factory floor. The system performs focal sweeps, aligns and fuses the images into an all-in-focus representation, and applies a dual CNN pipeline based on the YOLO architecture: one network detects and localizes holes, while the other classifies debris. All training images were collected with the prototype, ensuring consistent geometry and lighting. On a withheld test set from a proprietary ≈3700 image dataset of aerospace assets, HANNDI achieved per-class precision and recall near 95%. An end-to-end demonstration on representative aircraft parts yielded an effective task time of 13.6 s per hole. To our knowledge, this is the first handheld automated optical inspection system that combines mechanical enforcement of imaging geometry, controlled illumination, and embedded CNN inference, providing a practical path toward robust factory floor deployment.

## 1. Introduction

In the aerospace industry, both fighter jets and cargo aircraft often contain over one hundred thousand holes of varying diameters, depths, and materials. These may be rivet holes, machined holes, or fastener sites. During manufacturing, it is critical that FOd, whether generated during production or unintentionally introduced, does not go undetected. Even something as small as a dust bunny, if trapped in a hole and pushed into the aircraft when a fastener is installed, can cause catastrophic failures such as clogging a hydraulic pump.

With this risk in mind, aviation companies employ inspection practices to mitigate FOd. One widely implemented policy is the Clean As You Go (CAYG) protocol, which requires removing foreign objects throughout the build process. In practice, however, items are still missed, leaving FOd undetected.

As part of CAYG, inspections rely heavily on visual checks, with workers using their eyes and simple tools. This approach is inherently limited, error-prone, and subject to human fatigue, resulting in foreign objects often being overlooked.

Although some smart tools have been proposed, adoption remains limited. For example, the Workforce Augmenting Inspection Device (WAND) integrates a camera, lighting, and standoff wheel to capture consistent imagery of aircraft parts [1]. However, WAND lacks onboard intelligence, so images must still be reviewed manually or processed later. As such, it primarily improves documentation rather than actively assisting inspectors.

In hole inspections specifically, personnel often examine thousands of holes per day using mirrors, flashlights, or borescopes. Despite these aids, the task remains repetitive and fatiguing, leading to reduced thoroughness over long sessions.

The persistence of human error underscores the need for a true smart tool that can image a hole, automatically detect and localize it, and identify debris inside. Meeting these requirements demands both high-fidelity imaging and embedded intelligence for detection and classification.

The remainder of this work is organized as follows. Section 2 explores related work relevant to this task. Section 3 presents the materials and methods proposed in our approach. Section 4 reports results on a withheld test set and operational deployment. Finally, Section 5 provides concluding remarks.

## 2. Related Work

Various imaging techniques have been explored for our use case. Ultrasonic sensors such as Baumer’s Series 09 rely on time-of-flight measurements to detect objects through apertures as narrow as 3 mm, and their performance is unaffected by transparency, color, or surface finish [2]. These sensors are highly robust in difficult environments, but they provide only presence/absence detection, not the detailed imagery needed for FO characterization.

Infrared time-of-flight (ToF) systems capture synchronized depth and infrared intensity images, fuse them for improved segmentation, and classify materials with high accuracy [3]. While effective under varied lighting, ToF still lacks the fine detail required for small-scale FO analysis.

Broader infrared imaging can detect objects in low-light or degraded settings, which is useful in UAV and maritime surveillance [4]. However, many state-of-the-art small object methods rely on complex modeling and multi-stage processing, limiting their practicality for fast, high-fidelity FO inspection on the factory floor.

Standard high-resolution RGB cameras remain common due to their low cost and ease of integration. Yet their shallow depth of field inside cavities necessitates focal stacking to achieve full in-focus coverage.

Selecting the right imaging method is only part of the challenge. Effective FO detection also requires hole detection intelligence to localize the correct region for downstream analysis. Approaches range from classical image processing to modern deep-learning object detectors.

YOLOv3 has been applied in automotive and aerospace inspection. In automotive work, Sileo et al. [5] used YOLOv3 after 3D registration to refine peg-in-hole alignment. In aerospace, Meng et al. [6] applied it to isolate deep hole regions in composites, followed by edge enhancement and circle fitting for sub-0.03 mm accuracy. These examples highlight YOLOv3’s flexibility as both a final alignment tool and as an enabler of precise measurements in visually complex conditions.

Semantic segmentation has also been tested. Melki [7] compared Faster R-CNN with U-Net and LinkNet for detecting screw holes, finding segmentation to be more accurate, though limited to simulated, simple objects.

For embedded systems, lightweight models have emerged. Zhao et al. [8] proposed LPO-YOLOv5s, a MobileNetv3-based variant that reduced parameters by 45% with minimal mAP loss for pouring hole detection on resource-limited hardware.

Once the hole is located, FO classification is needed to identify debris with enough precision to avoid false positives in safety-critical contexts. Mirzaei et al. [9] reviewed small object detection challenges relevant to FO classification such as low resolution, poor feature representation, occlusion, and class imbalance. Wahyudi et al. [10] surveyed deep learning strategies, noting that while the accuracy improves, the computational load and inference time increase, which matter for embedded systems.

Building on these, Liu et al. [11] introduced ESOD, which filters the background early to focus computation on likely regions. Jing et al. [12] enhanced YOLOF with feature fusion, attention, and dilated convolutions, improving robustness without major parameter costs.

Hybrid methods also exist. Kos et al. [13] combined segmentation and tracking with a YOLOv7 tiny detector plus overlapping box suppression, improving accuracy while keeping computation low.

Some work has tailored YOLO specifically for small objects. Li et al. [14] improved YOLOv7 for traffic sign detection with added prediction scales, hybrid convolutions, and robust loss functions. Li et al. [15] modified YOLOv8 for tiny defect detection in cylindrical micro-holes, outperforming Faster R-CNN, YOLOv5, and the baseline YOLOv8 while retaining real-time speed.

Although these methods advance imaging, hole detection, and small-object classification, none combine high-fidelity capture with real-time, embedded detection and classification specifically for FO inspection in aerospace manufacturing. This gap motivated the development of HANNDI, which integrates controlled imaging and onboard intelligence for both hole localization and FOd classification.

## 3. Materials and Methods

### 3.1. Computer Vision Pipeline

The image capture process in HANNDI is designed to produce a single, high-quality, all-in-focus image of a hole’s interior for downstream processing. An Arducam 64 MP HawkEye camera is connected to a Raspberry Pi 5, providing high-resolution imaging with full software control over lens position. A NeoPixel LED light ring is mounted concentrically around the lens to ensure consistent illumination of the inspection area regardless of ambient lighting conditions. The LEDs are activated immediately prior to image capture and turned off afterward to conserve power and prevent glare.

The design of HANNDI enforces a minimum standoff distance of 3.15″ (8.0 cm), corresponding to the HawkEye’s closest in-focus range. With the camera head geometry fixed, the effective detection range extends from the hole entrance down to a depth of approximately 0.75″ (1.9 cm). To cover this range, the system captures a focal sweep of sixteen images at incrementally increasing lens positions. Features beyond this depth, such as through hole backgrounds, remain out of focus and are suppressed in the stacked image.

Although all experiments used the HawkEye module, the pipeline is generalizable to other cameras. Using a different lens or sensor would require recalibration of the minimum in-focus distance and an adjustment of the focal sweep increments, but the registration, stacking, and CNN inference stages remain unchanged.

The camera automatically handles exposure and white balance via the PiCamera2 driver. Because each focal sweep is illuminated entirely by the fixed-intensity LED ring, the auto exposure/white balance routines converge quickly and produce stable results. Preliminary tests showed that locking these parameters provided no measurable improvement, so the simpler auto adjust mode was adopted.

Each frame is saved at 1280×720 resolution to balance processing efficiency and image detail. Prior to stacking, frames are registered using ORB feature matching with affine warping. Registration accuracy was quantified using the root-mean-square centroid radius, which measures the pixel spread of keypoint centroids across the focal sweep. In two representative sweeps, this metric ranged from 60 to 70 px before alignment and was reduced to 1–2 px after alignment, effectively eliminating ghosting and preserving fine detail as seen in Figure 1. This step is critical in handheld use, where small operator motions can otherwise introduce misalignments between slices.

Following alignment, the sixteen RGB images are fused into a single all-in-focus image via focal stacking. Each frame is converted to grayscale, and a variance-based sharpness map is computed to highlight areas of high local detail. The sharpness maps are smoothed to reduce noise, and for each pixel location, the frame with the highest sharpness value is selected. The corresponding RGB pixel is then inserted into the output image. This process not only brings all surfaces within the 0.75″ depth into focus but also suppresses through-hole clutter, since distant backgrounds remain blurred and are excluded during pixel selection.

The stacked image is then cropped to a 640×640 pixel region centered on the hole. This cropping reduces background clutter while preserving sufficient resolution for accurate analysis. The resulting image serves as the input to the Hole Identifier convolutional neural network described later, which detects and localizes the hole for subsequent foreign object classification. The combination of controlled lighting, fixed standoff distance, focal stepping, feature aligned stacking, and targeted cropping ensures that the network consistently receives high-quality, information-rich inputs suitable for robust inference under realistic handheld operating conditions. An example final image generated from this pipeline can be seen in Figure 2.

To justify the choice of 16 focal steps, a trade-off study compared 12, 16, and 20 slices. Median Laplacian variance, a measurement of image sharpness, of the final stacked image improved from ≈425 at 12 steps to ≈435 at 16 steps, with only marginal gains up to ≈445 at 20 steps. Median runtime for image acquisition, feature matching, and stacking scaled nearly linearly, with 5.9 s at 12 steps, 7.1 s at 16 steps, and 8.7 s at 20 steps. Sixteen steps provided the best balance of sharpness and runtime, while also avoiding the variability observed at 20 steps. Moreover, Lockheed Martin defined a target of sub-10 s for the scanning stage, and while 20 steps at 8.7 s left little margin for hole detection and FOd classification, the 7.1 s runtime at 16 steps provided comfortable headroom. These factors together motivated the selection of 16 steps as the operational setting, as shown in Figure 3.

### 3.2. Physical System Design

The HANNDI device is designed for portable, real-world deployment, with an emphasis on reliable operation, ergonomic use, and a clean integration of all imaging and control components. At the core of the power system is a 7.4 V, 5000 mAh, 50 C LiPo battery that enables untethered operation. A toggle switch is mounted inline to allow the user to fully power down the system between uses, conserving battery life. A 5 A inline fuse is included to protect downstream hardware in the event of a short or sudden current dump. Power is regulated through a buck converter that steps the voltage down to a stable 5 V, supplying up to 5 A to ensure the Raspberry Pi 5 and its peripherals receive consistent and adequate power during operation.

The Raspberry Pi 5 serves as the central controller, managing both image processing and peripheral communication. Connected peripherals include a pushbutton for initiating image capture, a NeoPixel LED light ring for consistent illumination, an Arducam 64 MP camera for high-resolution imaging, and an LCD screen for providing real-time feedback to the user. Together, these components draw moderate current, and under typical load conditions, the system achieves approximately six hours of continuous operation per full battery charge.

All electrical and mechanical components are housed in a purpose-built smart tool that supports field use by manufacturing personnel. The CAD model design for HANNDI is shown in Figure 4. The handle section houses the LiPo battery, power switch, inline fuse, buck converter, and image capture button. Above the handle is an electronics enclosure that contains the Raspberry Pi 5, internal wiring, and the LCD display mounted for easy visibility during operation.

Mounted to the front of the electronics box is the camera head, which houses both the Arducam camera and the LED ring light. The head is mechanically connected via a spring that allows for positional adjustment, enabling users to manually rotate the camera to align it with a hole or fastener site of interest. This feature is especially important when working on complex geometries or hard-to-reach areas.

The camera head encloses both the camera and LED ring in a barrel-like geometry, leaving only a 0.25″ (6.3 mm) mechanical gap between the head and the inspected surface. This gap is due to the compression limits of the shocks in the suspension mechanism. As a result, nearly all ambient light is excluded from the inspection region, with illumination dominated by the integrated LED ring. The suspension design, combined with the fixed 3.15″ (8.0 cm) standoff enforced by the camera head geometry, ensures that the lens axis remains parallel to the inspected surface, including on gently curved aerospace components. This compliance preserves the required standoff distance across sloped or contoured features, keeping the imaging geometry consistent across parts.

The full device measures approximately 18″ (46 cm) in length when the camera head is not rotated. It is intended for two-handed operation, with one hand gripping the handle for control and the other adjusting the camera head for alignment. This combination of stability and adjustability allows for accurate image capture in a wide range of aerospace inspection environments. The completed prototype can be seen in Figure 5, where the adjustable camera head and suspension mechanism is also shown in action on a curved surface.

For hole illumination, the system uses a concentric NeoPixel RGBW LED ring with 12 integrated 5050 LEDs arranged at a 36.8 mm outer diameter and 23.3 mm inner diameter. Each LED includes 8-bit PWM control of the R, G, B, and W channels. In this work, equal RGB values of (5, 5, 5) were applied, corresponding to a roughly 2% duty cycle per channel. This provided sufficient brightness for inspection without saturating the sensor.

Illuminance tests confirmed that the LED ring overwhelmingly dominated residual ambient factory lighting. In a test environment, under only ambient illumination, the hole surface measured about 455 lux. With the camera head in place and LEDs off, this dropped below 5 lux due to the head blocking stray light. With the LEDs active, the level rose to approximately 300 lux. This corresponds to a residual ambient-to-flash ratio greater than 60:1, since the camera head effectively reduced ambient light at the surface to negligible levels while the LEDs provided controlled illumination. As a result, the illuminance in captured images was determined almost entirely by the LED system, independent of external lighting variability.

### 3.3. CNN Methodology and Training

The HANNDI device uses two CNNs based on the YOLOv7-Tiny architecture. A primary motivation of this work was the ability to deploy a single, unified backbone across both pipeline stages. YOLOv7-Tiny could be used without modification as a detector for the Hole Identifier CNN and, in single-bounding-box mode, as a classifier for the FOd Classifier CNN [16]. By contrast, adopting base CNNs such as MobileNet or EfficientNet would have required using one variant for classification and a modified version such as MobileNet-SSD or EfficientDet for detection. That approach would have introduced additional training procedures and integration overhead. The ability to reuse a single, unaltered YOLO model streamlined implementation and reduced complexity on the embedded platform.

YOLOv7-Tiny also delivered strong out-of-the-box performance on both hole detection and FOd classification. Prior literature shows that MobileNet-SSD and EfficientDet variants achieve broadly comparable accuracy on small-object detection, but not clear gains [17,18]. A similar conclusion can be drawn for the low probability of obtaining significant improvements with base MobileNet or EfficientNet classifiers. Given the strong baseline achieved by our trained CNNs, as discussed later in this subsection, additional benchmarking against alternative backbones was unlikely to provide meaningful benefits in the present context.

The architecture further provides extensibility and practical embedded performance. Although the current FOd stage performs classification only, the same YOLO framework can readily be extended to localize debris within holes, enabling future transitions from classification to detection with minimal modification. By contrast, an EfficientNet based classifier would require a switch to a different architecture such as EfficientDet, complicating deployment.

The model is also designed for low-latency inference and runs natively on lightweight hardware such as the Raspberry Pi 5. While EfficientDet and MobileNet-SSD have been reported to achieve slightly faster inference times [19], these differences are negligible in our context. Image acquisition, registration, and stacking require approximately seven seconds per scan as discussed in Section 3.1, whereas CNN inference is on the order of milliseconds. We can therefore conclude that the total scan time is dominated by operations other than CNN inference, making any minor gains in inference speed irrelevant.

Moreover, the novelty of this work lies not in CNN benchmarking, but in the design, integration, and deployment of a unified inspection system for aerospace use on resource-constrained embedded hardware. This study demonstrates that a single lightweight object detector can serve dual roles, both detection and classification, without compromising accuracy, enabling real-time operation within a handheld inspection device.

Having established the rationale for the YOLO backbone in our CNN pipeline, we now describe the dataset and training procedures used for both models. All training images were collected directly with the HANNDI prototype on Lockheed Martin aerospace assets. This ensured consistent imaging geometry and illumination between training and deployment. The proprietary dataset contained about 3700 images. Approximately 400 images did not contain any hole. The remaining 3300 images contained holes staged on parts in a simulated aerospace manufacturing environment. Some of these holes were intentionally left clean. The rest contained one of five FOd types, which were blue masking tape, metallic burrs, composite shreds, sealant flakes, and dust bunnies.

In the CNN pipeline, two YOLOv7-Tiny models were used. The first, the Hole Identifier CNN, detects and localizes holes in 640 × 640 stacked images. If no hole is detected, the pipeline terminates. If a hole is found, the region is cropped to 320 × 320 pixels and passed to the second model, the FOd Classifier CNN, which determines whether the hole is clean or contains debris.

For the Hole Identifier CNN, the full dataset of 3700 images was split 80–20% into training and validation sets. The network was trained for 30 epochs using standard YOLOv7 procedures. Validation metrics initially rose rapidly and stabilized after approximately 20 epochs. At convergence, the model consistently achieved >95% precision and recall, with mAP@0.5 ≈ 0.98. Training and validation loss curves decreased smoothly with no signs of overfitting, with final validation losses stabilizing at around 0.03. These results, as shown in Figure 6 and Figure 7, confirm accurate and stable hole localization across varied imagery.

The 3300 hole images were cropped using the Hole Identifier CNN to form the 320 × 320 dataset for the FOd Classifier CNN. This dataset was split into 70% training, 20% validation, and 10% testing. The network was trained for 75 epochs, with validation metrics plateauing after 60 epochs. Final validation performance reached 95% precision and recall across all six classes. Training and validation loss curves tracked closely, with final validation losses stabilizing at around 0.01, again showing no evidence of overfitting. The 10% test set was held out for independent evaluation, which is discussed later. Training dynamics are shown in Figure 8 and Figure 9.

## 4. Results

### 4.1. Test Performance

To evaluate the CNN pipeline’s performance, we begin by presenting two representative outputs from the Hole Identifier CNN in Figure 10 using images from the training set of 3700 images. In the first example, the model correctly detects a hole and draws a bounding box around it. In the second, where no hole is present, the model outputs no hole detected. These results highlight the model’s ability to distinguish between valid and invalid hole detection cases.

The FOd Classifier CNN was then tested on the withheld set of cropped hole images that were never seen during training or validation. The model achieved an overall precision and recall of approximately 94% when evaluated across all FOd classes and clean holes. Among the classes, the composite showed a slightly lower recall at around 86%, which means that 14% of composite images were misclassified as other FOd types or as clean holes. Metallic burr and dust bunny showed the lowest precision, each near 90%, meaning that about 10% of the images predicted as these classes actually belonged to different categories. Despite these few outliers, the overall classification performance remained strong. The complete table of metrics obtained from this test is shown in Table 1.

To better visualize the FOd classifier’s performance, a confusion matrix was generated using the test set predictions as seen in Figure 11. In this matrix, the horizontal axis represents the true label while the vertical axis shows the predicted label. Diagonal values close to 1 indicate perfect classification. Blue tape was classified perfectly, with a diagonal value of 1. Sealant and clean hole were also classified with high accuracy. The model struggled the most with the composite, metallic burr, and dust bunny classes. For instance, 17% of composite samples were predicted as metallic burr, and 12% of metallic burr samples were misclassified as composite. Dust bunny classifications were more scattered, with clean hole being the most common incorrect prediction. These results suggest that the model occasionally confuses similar FOd types, especially composite and metallic burr, but such errors were relatively limited, which is reflected in the high overall precision and recall.

To further characterize classifier performance across thresholds, Precision–Recall (PR) curves were generated for all six output classes in Figure 12. The PR curves yielded average precision (AP) values ranging from 0.954 to 0.996. Blue tape, sealant, and dust bunny exhibited near-perfect curves, indicating consistent classification across thresholds. Clean hole also showed robust performance with an AP near 0.98. Composite and metallic burr lagged slightly behind, with AP values of 0.954 and 0.966, respectively, confirming these categories as the most challenging to separate. Together, the precision and recall values, confusion matrix, and PR curves provide a comprehensive view of test performance, confirming both the overall robustness of the classifier and the specific classes most prone to confusion.

To demonstrate the full CNN pipeline, two image examples are provided in Figure 13. A 640 × 640 image generated by the computer vision pipeline is first processed by the Hole Identifier CNN. If a hole is detected, the image is cropped based on the bounding box to a 320 × 320 region centered on the hole. This cropped image is then passed to the FOd Classifier CNN, which returns the predicted class along with a confidence score. One of the examples shows a correctly classified clean hole, while the other contains a correctly classified metallic burr.

While overall classification accuracy was strong, example misclassifications are shown in Figure 14. In the first example, a metallic burr was predicted as a dust bunny due to dim illumination reducing the specular highlights that normally differentiate the two. Similarly, composite shreds in the second example were labeled as a metallic burr due to overlapping silhouettes and edge structure. These errors reflect both illumination sensitivity and the intrinsic shape and texture similarities between certain FOd types. The dataset size, approximately 500 images per FOd class and 800 images of clean holes, also likely contributed to these misclassifications by limiting the diversity of training examples. Future work will address these limitations through expanded data collection and potential refinements to the camera head lighting geometry and multi-slice stacking strategy to further improve illumination and enhance discriminative detail.

### 4.2. Operational Deployment

During pipeline development, a confidence threshold was introduced to reduce false positive hole detections and low-certainty FOd predictions. Hole detections with a confidence below 75% are flagged as “No Hole in Frame”, and FOd classifications with a confidence below 50% are flagged as “Invalid Class”. In practice, this means that only classifications with a confidence ≥ 0.50 are reported as valid, while lower confidence outcomes are explicitly labeled as invalid to avoid forcing a guess. This threshold was selected as a majority confidence criterion, balancing sensitivity to true debris with conservatism against false positives, and providing a clear safeguard against the misclassification of unknown debris.

The trained model weights for both neural networks, along with the logic for image cropping and confidence filtering, were integrated with the computer vision pipeline on the Raspberry Pi 5. This finalized the HANNDI software system. The prototype was then able to capture images across sixteen focal planes with uniform lighting, align and stack them into a single all-in-focus image, and crop it to 640 × 640 pixels for hole detection. If a hole is detected by the Hole Identifier CNN, the system crops the image again to 320 × 320 based on the bounding box, then runs the cropped image through the FOd Classifier CNN to determine the type of debris or confirm that the hole is clean. At each stage, the LCD screen displays a status update such as capturing, stacking, detecting, and the final result. The final output is either the predicted class with a confidence value, an “Invalid Class” message if no prediction exceeded 0.50 confidence, or a notification that no hole was present in the stacked image.

An end-to-end demonstration was then performed on Lockheed Martin aerospace assets, including a flat panel with blind holes, a one-eighth fuselage model with blind holes on contoured surfaces, and a wing panel with through holes. These assets presented different surface conditions ranging from white and dull metallic to glossy grey and dark metallic. A scripted circuit of seven inspection sites was executed in sequence. During this demonstration, the system produced correct outputs in every case, with predicted classes and confidence values summarized in Table 2.

The full demonstration required 95 s to complete seven scans, yielding an effective task time of approximately 13.6 s per hole. This figure includes operator handling, focal sweep capture, CNN inference, and LCD feedback. This runtime falls comfortably within the operational target of 15 s identified by Lockheed Martin quality engineers.

Due to Lockheed Martin restrictions, neither the CAD geometry of the HANNDI device nor the raw image dataset can be released. However, all hardware specifications, illumination geometry, and CNN training configurations are described in sufficient detail to support reproducibility by qualified researchers. Interested collaborators may request access from the corresponding author. Subsequent access is subject to Lockheed Martin standard review and approval procedures.

## 5. Conclusions

This work introduces HANNDI, a handheld automated optical inspection tool that enforces a fixed standoff and controlled illumination for consistent image quality. A dual-CNN pipeline based on YOLOv7-Tiny was trained on ≈3700 proprietary images collected with the prototype, enabling reliable hole detection and FOd classification. Across test evaluations, the system achieved ≈95% precision and recall while maintaining an effective task time of 13.6 s per hole.

In contrast to earlier efforts such as the Workforce Augmenting Inspection Device (WAND), which focused on improving image capture without embedded intelligence, HANNDI integrates controlled optics, mechanical compliance, and deep learning inference within a portable device. This combination provides both consistent image quality and immediate classification on the factory floor, addressing a critical gap in current inspection workflows.

Future work will expand the dataset with more diverse real world examples, validate performance across different environments, benchmark YOLOv7-Tiny against other CNN backbones to evaluate potential trade-offs, and conduct operator studies to assess usability and ergonomics, including potential redesigns of the handle and camera head for improved comfort. Together, these efforts will further strengthen HANNDI as a foundation for intelligent inspection in aerospace manufacturing.

## Figures and Tables

**Figure 1 sensors-25-05921-f001:**
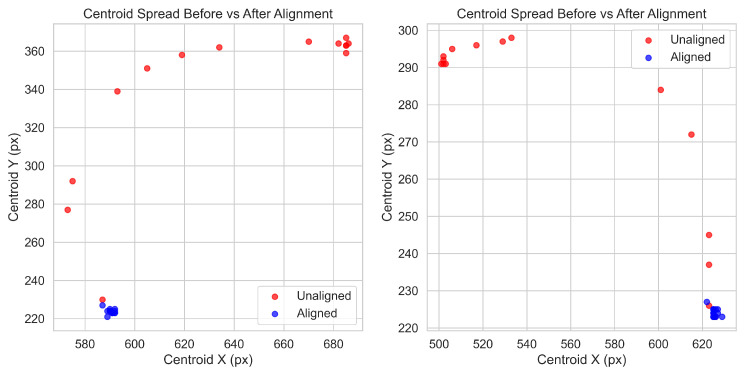
Hole centroid registration examples.

**Figure 2 sensors-25-05921-f002:**
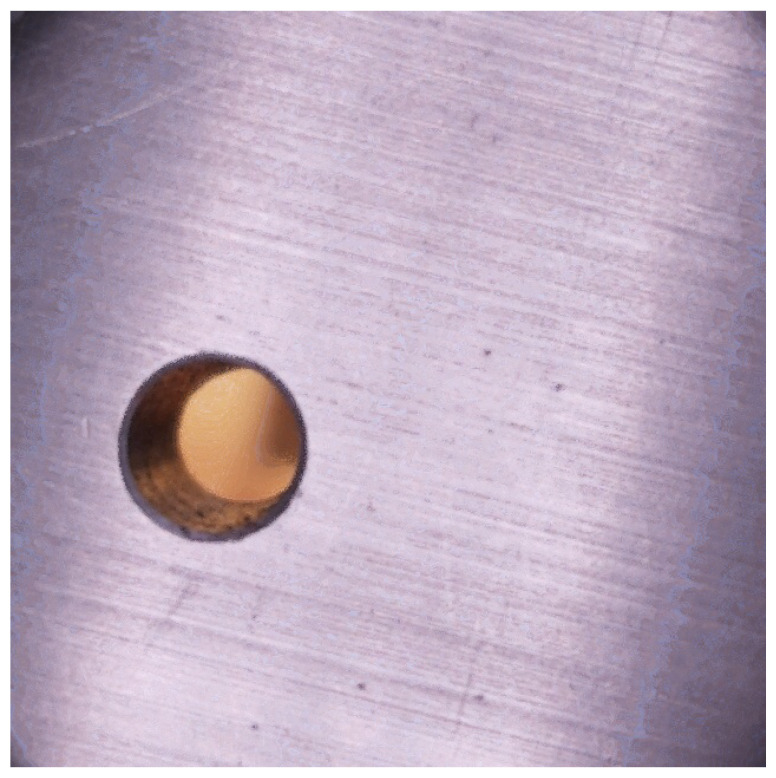
Computer vision pipeline image example.

**Figure 3 sensors-25-05921-f003:**
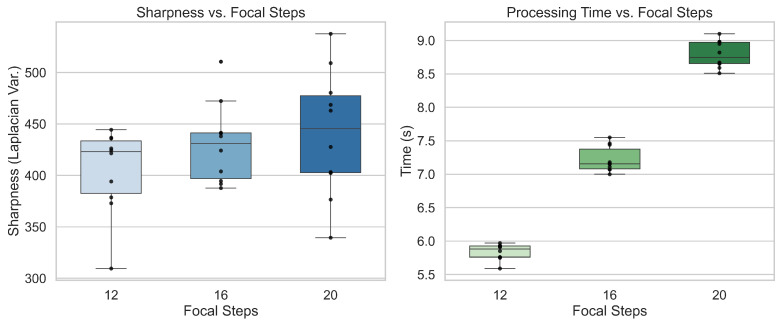
Image sharpness and processing time for 12, 16, and 20 focal steps.

**Figure 4 sensors-25-05921-f004:**
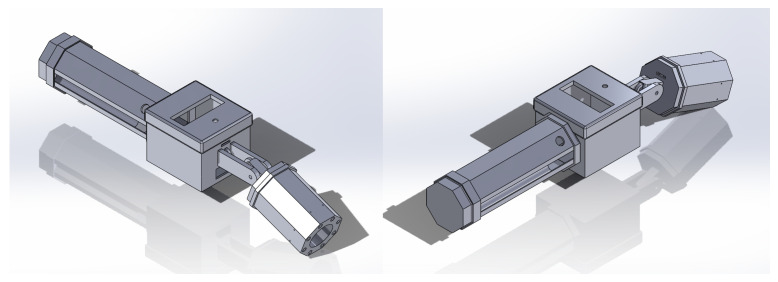
HANNDI CAD model isometric views.

**Figure 5 sensors-25-05921-f005:**
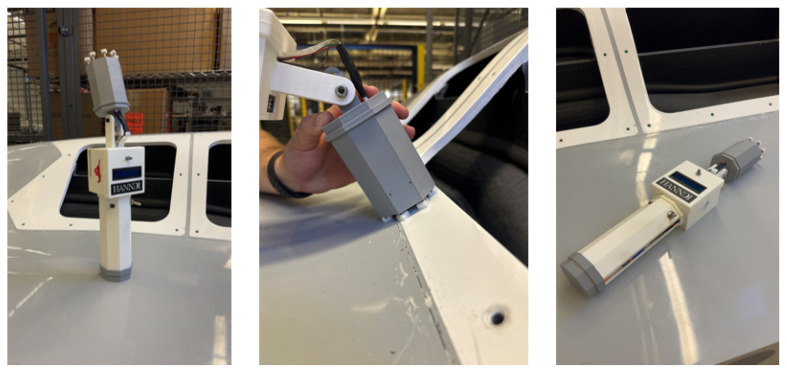
HANNDI prototype; close up of suspension mechanism on contoured surface.

**Figure 6 sensors-25-05921-f006:**
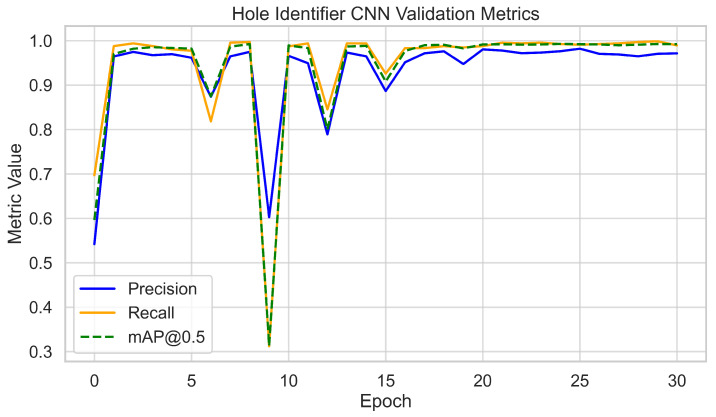
Hole Identifier CNN validation metrics per epoch (precision, recall, and mAP@0.5).

**Figure 7 sensors-25-05921-f007:**
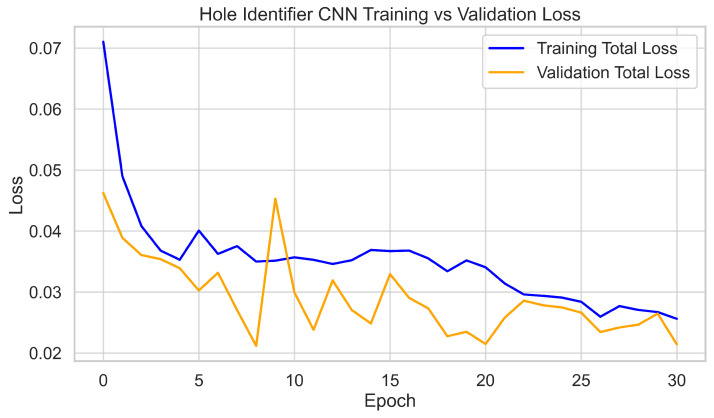
Hole Identifier CNN training and validation loss curves.

**Figure 8 sensors-25-05921-f008:**
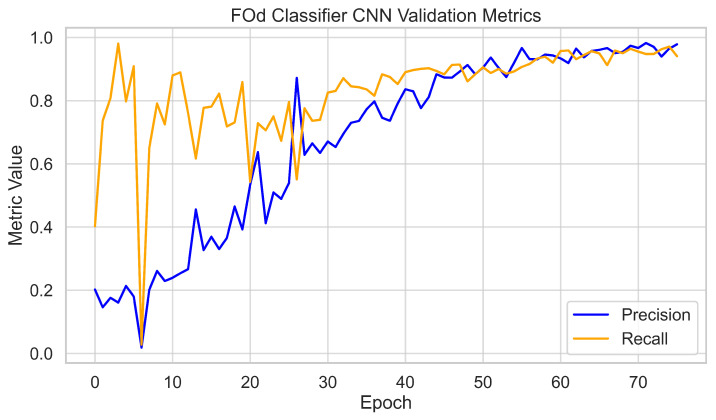
FOd Classifier CNN validation metrics per epoch (precision and recall).

**Figure 9 sensors-25-05921-f009:**
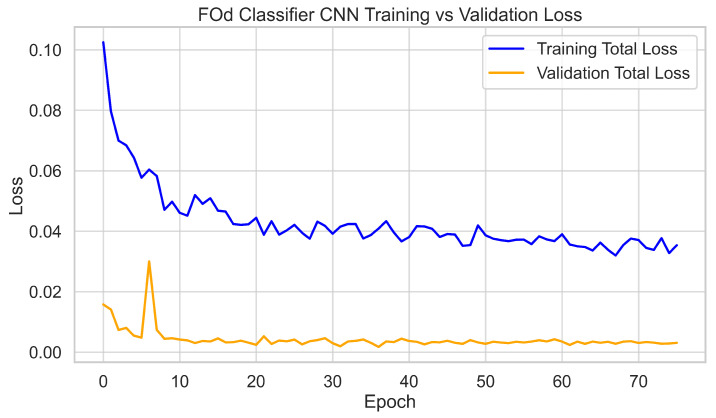
FOd Classifier CNN training and validation loss curves.

**Figure 10 sensors-25-05921-f010:**
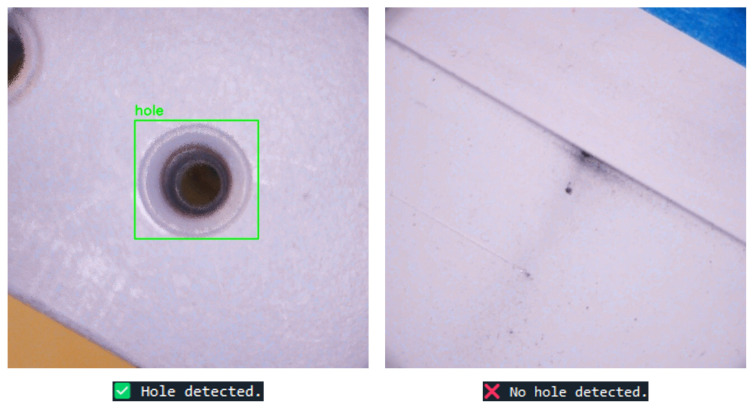
Hole Identifier CNN example output.

**Figure 11 sensors-25-05921-f011:**
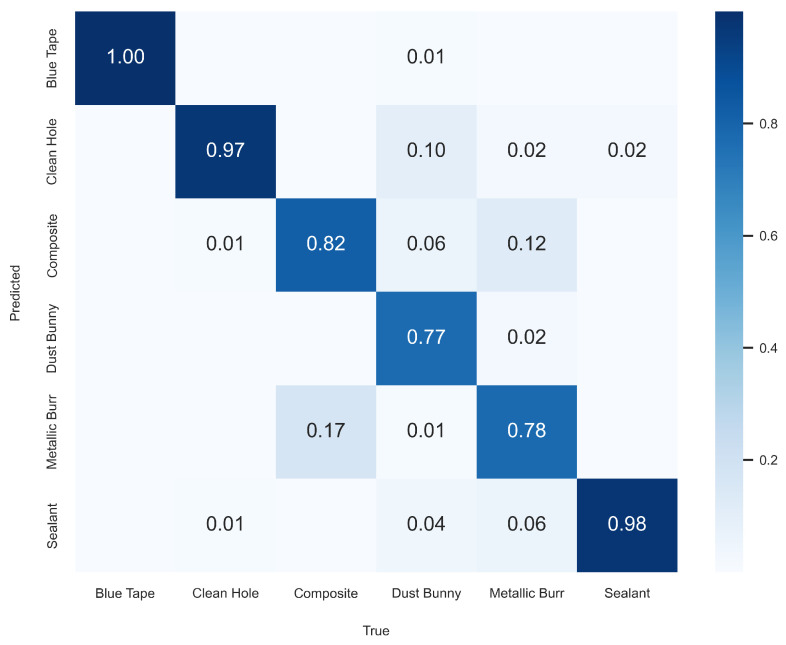
FOd Classifier CNN test set confusion matrix.

**Figure 12 sensors-25-05921-f012:**
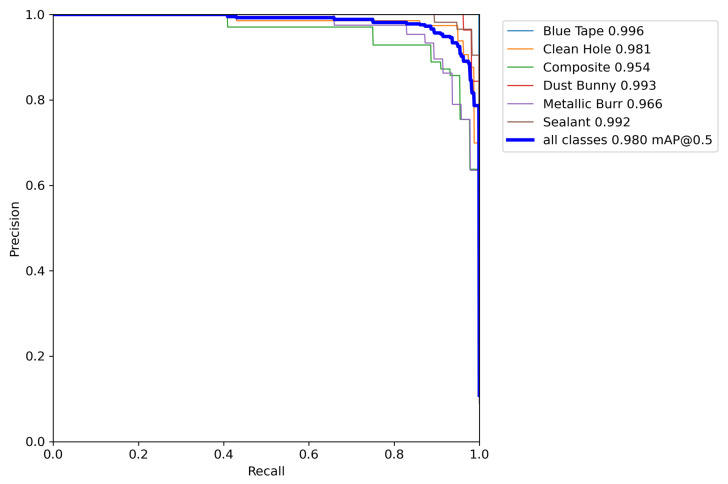
Precision–Recall (PR) curves for the FOd Classifier CNN on the test set.

**Figure 13 sensors-25-05921-f013:**
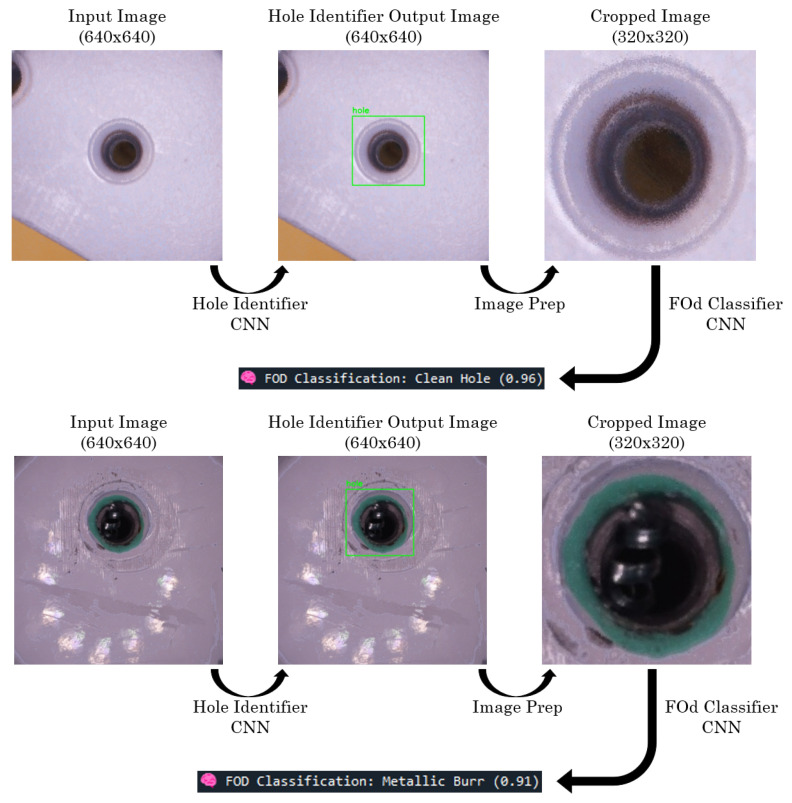
CNN pipeline correct classification examples.

**Figure 14 sensors-25-05921-f014:**
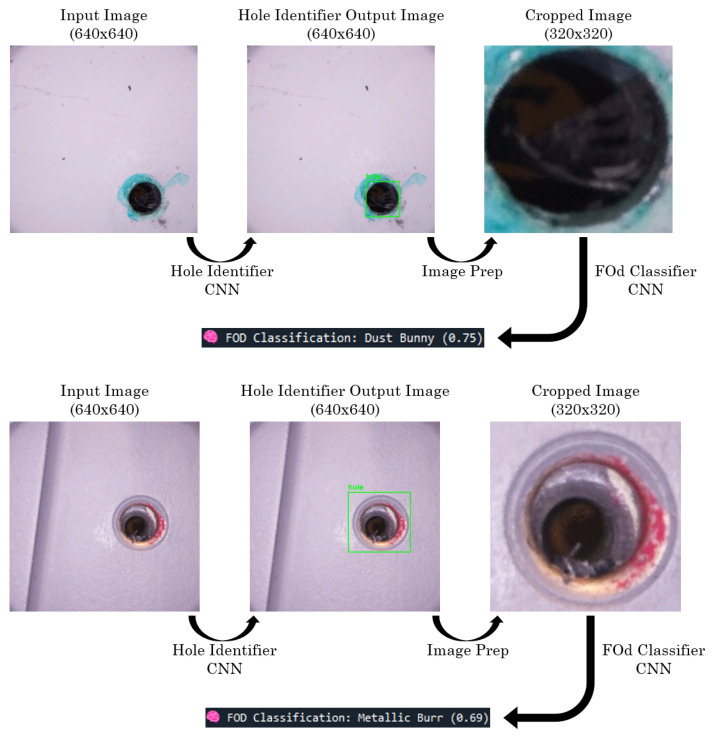
CNN pipeline misclassified examples.

**Table 1 sensors-25-05921-t001:** Per-class precision (P) and recall (R) for the FOd classifier CNN on test set.

Class	Labels	P	R
All	337	0.941	0.940
Blue Tape	56	0.989	1.000
Clean Hole	79	0.973	0.926
Composite	44	0.927	0.864
Dust Bunny	54	0.900	0.981
Metallic Burr	47	0.895	0.915
Sealant	57	0.964	0.951

**Table 2 sensors-25-05921-t002:** Results of HANNDI demonstration run across Lockheed Martin assets.

Asset	Hole	Predicted Class	Confidence
Flat Panel	Hole #1	Clean Hole	0.76
	Control	No Hole in Frame	–
	Hole #2	Blue Tape	0.94
Fuselage Model	Hole #1	Composite Shred	0.90
	Hole #2	Metallic Burr	0.86
Wing Section	Hole #1	Dust Bunny	0.97
	Hole #2	Sealant Flake	0.72

## Data Availability

The data used in this study were collected at Southern Methodist University and are not publicly available due to institutional restrictions. However, they are available from the corresponding author upon reasonable request.

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
