# Peer review of "Convolutional Neural Networks for Hole Inspection in Aerospace Systems"

_sensors, 2025, doi:10.3390/s25185921_

Round 1
Reviewer 1 Report
Comments and Suggestions for Authors
- Were all training images captured with your custom handheld device? If not, quantify how capture angle, standoff, and pose affect model performance. What are the valid detection range and angles for reliable inference? Please provide performance vs. angle/range curves. If a camera with different optical specifications (lens focal length/f-number, sensor size/pixel pitch, distortion/MTF) is used, how does performance change? Please include cross-camera generalization tests and any required recalibration.
- Do you require strictly controlled lighting, or can the network infer reliably under typical factory variability in spectrum, intensity, and glare? What is the handling performance across representative lighting conditions, including spectrum, intensity and glare? Please also describe and justify all controls: exposure/gain locks, white balance lock… etc. Provide before/after results to show the incremental benefit of each control if there is any.
- Please pecify ring-light geometry and output (luminous flux, beam pattern, diffuser… etc.), driver characteristics, and the ambient-to-flash ratio during capture. Please also show that ambient variations (lights on/off, sun ingress) do not materially affect predictions, or define the acceptable ambient-light tolerance window.
- How do you define “unknown” debris operationally? When an instance is not in your predefined classes, does the system output Unknown, Invalid, Clean, or the nearest known class? Please specify the exact decision logic.
- Please demonstrate end-to-end performance on a realistic factory panel with N holes and state the target takt time for the intended production context and show whether your system meets it, along with concrete optimizations and their measured impact.
- Why 16 focal steps? Please show an accuracy–latency trade-off across other numbers and defend the chosen operating point. Please also specify the registration algorithm, quantify residual alignment error and its impact on downstream detection/classification.
- Please explain handling of curved/tilted surfaces and through-holes where the background comes into focus, including any depth gating, deblending/masking strategies, and documented failure cases with mitigations.
Reviewer 2 Report
Comments and Suggestions for Authors
The manuscript presents a well-motivated and experimentally thorough approach to automated optical inspection (AOI) using a combination of classical machine learning and deep convolutional neural networks (DCNNs). The hybrid model design and the use of both proprietary and public datasets demonstrates practical relevance. However, despite strong experimental coverage, several critical issues must be addressed before publication:
- Lack of Visual Diagnostic Tools
- What’s Missing: No confusion matrices, training/validation accuracy/loss curves, or PR curves are provided for all tested models.
- Why It Matters: These are critical for verifying whether the models are overfitting, biased toward the majority class, or misclassifying rare defect types.
- What Require:
Add confusion matrices for at least the best-performing models on each dataset (full, augmented, proprietary).
Provide training/validation loss and accuracy curves to assess convergence.
Include PR curves.
- 2. No Error Analysis
- Issue: No qualitative or case-wise error analysis is presented (e.g., false positives/negatives on difficult cases).
What Require:
Include examples of misclassified images and provide insights into why the model failed (e.g., ambiguous features, poor lighting, data overlap, susceptibility to light reflections).
- 3. Language
- Issue: Give examples of incorrectly classified images together with an explanation of the model's weaknesses (e.g., ambiguous features, inadequate illumination, data overlap).
What Require:
A thorough language revision for fluency and conciseness (especially in the abstract, introduction, and methodology).
How to Reproduce Results?
4: Also share how to regenerate results for verification.
5: Very nice handheld demo (HANNDI) but lack of user experience testing and CAD mechanical and electronic files for reproduction of device.
Reviewer 3 Report
Comments and Suggestions for Authors
This paper presents the Handheld Automated Neural Network Detection Instrument (HANNDI), a device based on a dual-CNN YOLO network model for detecting foreign object debris in rivet holes, machined holes, and fastener sites on aerospace components. The work is interesting; however, some adjustments are required, as outlined below:
* The introduction is too long and could be divided into Introduction and Related Work. In the Introduction section, the problem should be presented by moving from the general to the specific, outlining the problem statement, justification, objectives, and a description of the rest of the paper. The Related Work section should be focused on appropriately describing the state of the art.
* A stronger justification for choosing the YOLO network is needed. While YOLO is indeed a popular CNN that can be implemented in real time on embedded systems, there are many other networks with similar characteristics, such as MobileNet, SqueezeNet, EfficientNet, and Convolutional Transformers, among others.
* The graphs in Figures 4 and 5 require clearer descriptions. It is not evident how the training or validation sets behave. Furthermore, a detailed description of the network’s performance based on the behavior of the evaluation metrics should be provided.
* The content of Figure 7 should be presented in the form of a table.
*The captions within the confusion matrix in Figure 8 are of low resolution. While this may be due to Python’s default settings, the captions can and should be improved.
*An important aspect of methods based on deep learning or neural networks is the comparison with other approaches. The results section should include a comparison of the proposed model with other deep networks based on CNNs or transformers. Without a comparative analysis showing that the proposed model performs better than traditional deep learning models for defect detection, the work may be interesting as an engineering project but does not fall within the frontier of scientific knowledge.
Round 2
Reviewer 1 Report
Comments and Suggestions for Authors
Thank you very much for your response. The authors have provided a comprehensive and clear response to every point raised. Their revisions have substantially improved the manuscript, particularly by adding the crucial descriptions and refining the figures to a publishable standard. The new data also effectively strengthens their core claims and provides a much clearer picture for the reader.
Author Response
We sincerely thank the reviewer for their positive evaluation of our revised manuscript. We are pleased that the additional data, figures, and clarifications strengthened the core contributions and improved readability. We appreciate the reviewer’s constructive input during the first round of review, which substantially improved the final version of this work.
Reviewer 3 Report
Comments and Suggestions for Authors
Thank you, as many of my observations were taken into account. However, there are still some details to consider for the following reasons:
While the YOLO family of models was designed for object detection, base CNN models such as MobileNet, EfficientNet, and SqueezeNet are typically used as classifiers when dense layers and a sigmoid/softmax function are added in the final dense layer. If deconvolutional layers are placed after the base CNN model, the resulting architecture can be used for detection, segmentation, and other tasks. For example, see the following works:
*Real-Time Object Detection Using SSD MobileNet Model of Machine Learning
*Overview of Object Detection Algorithms Using Convolutional Neural Networks
*Object Detection Using Deep Learning, CNNs and Vision Transformers: A Review
These articles present other CNN-based models for object detection tasks. Therefore, I believe a different justification should be provided for choosing YOLO, and it would be interesting to include a performance study comparing YOLO against at least two different base CNNs.
